# Optimizing the Preoperative Preparation of Sarcopenic Older People: The Role of Prehabilitation and Nutritional Supplementation before Knee Arthroplasty

**DOI:** 10.3390/nu16203462

**Published:** 2024-10-12

**Authors:** Francesco Pegreffi, Rita Chiaramonte, Sabrina Donati Zeppa, Fulvio Lauretani, Marco Salvi, Irene Zucchini, Nicola Veronese, Michele Vecchio, Alessia Bartolacci, Vilberto Stocchi, Marcello Maggio

**Affiliations:** 1Department of Medicine and Surgery, School of Medicine and Surgery, “Kore” University of Enna, 94100 Enna, Italy; francesco.pegreffi@unikore.it; 2Unit of Recovery and Functional Rehabilitation, P. Osp. Umberto I, 94100 Enna, Italy; 3Unit of Disability, Handicap, Territorial Rehabilitation, and Prosthetic Assistance, Azienda, Sanitaria Provinciale (ASP), 95124 Catania, Italy; ritachiaramd@gmail.com; 4Department of Biomedical and Biotechnological Sciences, University of Catania, 95124 Catania, Italy; michele.vecchio@unict.it; 5Department of Biomolecular Science, University of Urbino Carlo Bo, 61029 Urbino, Italy; a.bartolacci2@campus.uniurb.it; 6Geriatric Clinic Unit, University Hospital of Parma, Via Gramsci 14, 43126 Parma, Italy; fulvio.lauretani@unipr.it (F.L.); marco.salvi@unipr.it (M.S.); irene.zucchini@unipr.it (I.Z.); marcellogiuseppe.maggio@unipr.it (M.M.); 7Department of Medicine and Surgery, University of Parma, Via Gramsci 14, 43126 Parma, Italy; 8Department of Internal Medicine, Geriatrics Section, University of Palermo, Piazza delle Cliniche 2, 90127 Palermo, Italy; nicola.veronese@unipa.it; 9Department of Human Sciences for the Promotion of Quality of Life, University San Raffaele, 20132 Rome, Italy; vilberto.stocchi@uniroma5.it

**Keywords:** orthogeriatrics, sarcopenia, prehabilitation, dietary supplements, surgical resilience, knee arthroplasty

## Abstract

Background: Age-related loss of skeletal muscle strength and mass is linked to adverse postoperative outcomes in older individuals with sarcopenia. Half of patients suffer from severe associated osteoarthritis requiring orthopedic interventions. Mitigating the onset and progression of sarcopenia before surgery is essential to improve the prognosis and reduce surgical complications. The aim of this research was to innovatively explore whether the preoperative period could be the appropriate timeframe to empower surgical resilience, through prehabilitation and dietary supplementation, in older sarcopenic patients undergoing knee arthroplasty. Methods: The current literature concerning the effectiveness of prehabilitation and dietary supplementation before knee arthroplasty in sarcopenic older individuals was reviewed, following the SANRA criteria, between December 2023 and February 2024. The study inclusion criteria were as follows: (1) prehabilitation and/or dietary supplementation interventions; (2) human participants aged 65 years and older; (3) relevant outcome reporting (functional status, postoperative complications, and patient-reported outcomes); and (4) articles written in English The extracted information included study characteristics, demographics, intervention details, outcomes, and the main findings. Results: Merged prehabilitation and dietary supplementation strategies extrapolated from the current literature and involving strength, resistance, balance, and flexibility training, as well as essential amino acids, iron, vitamin D, adenosine triphosphate, and glucosamine sulphate supplementation, could improve the functional capacity, ability to withstand the upcoming surgical stressors, and postoperative outcomes in older people undergoing knee arthroplasty. Conclusions: Addressing complex links between knee osteoarthritis and sarcopenia in older individuals undergoing knee arthroplasty requires a multidimensional approach. Prehabilitation emerges as a crucial preliminary step, allowing the optimization of surgical outcomes. Nutraceutical integration, included in a comprehensive care plan, could have a synergic effect in achieving prehabilitation goals. Those interventions are essential for surgical resilience, in terms of muscle function preservation, recovery acceleration, and overall quality of life enhancement. Intensive collaboration among specialists could advance knowledge and the sharable consensus concerning the critical and evolutive field of perioperative care.

## 1. Introduction

Sarcopenia, characterized by the age-related loss of skeletal muscle strength and mass, is linked to adverse outcomes such as falls, fractures, disability, and increased mortality in older individuals and hospitalized patients [1]. The prevalence of sarcopenia depends on region, age, and classification. It ranges from 1% to 29% in community-dwelling older people and from 14% to 33% in the long-term-care population, as well as standing at approximately at 10% in acute hospital care settings [2]. Of the older people with sarcopenia, half suffer from severe associated osteoarthritis requiring orthopedic interventions [3]. In vulnerable surgical patients, particularly those who are older, frail, and sarcopenic, the challenge of enduring surgery is significantly harder, with reduced chances of a successful and timely recovery. Notably, sarcopenia holds prognostic significance in surgical patients, acting as an unfavorable predictor of postoperative complications and overall negative outcomes. Thus, mitigating sarcopenia prior to surgery is essential for improving a patient’s prognosis and reducing complication risks [4].

Older and frail individuals often suffer from metabolic syndrome and other metabolic disorders. Preoperative metabolic disturbances, typically insulin resistance, could lead to patients having an increased level of postoperative catabolism [5]. Furthermore, together with the age-related decrease in skeletal muscle mass and function [6], the concomitant elevated postoperative muscle proteolysis [7] and the surgical trauma related to knee arthroplasty exacerbate the pre-existing muscle dysfunction, resulting in increased post-surgical muscle weakness and disability. In fact, recent studies suggest that the reduction in both strength and muscle activation occurs with an incidence of up to 60% and 17% immediately after total knee arthroplasty (TKA) [8]. The loss of strength after surgery highlights a significant neural inhibition, primarily driven by the reduced activation of the quadriceps muscle, a phenomenon called arthrogenic muscle inhibition (AMI) [9]. To improve the post-surgical recovery of quadriceps muscle function and facilitate long-term rehabilitation, it is imperative to substantially reduce AMI in the early postoperative phase, but also to reinforce muscle function before surgery. The presurgical period represents a window of opportunity to enhance the individual’s resilience, providing them with the ability to compensate for the post-surgical reduction in their physiological reserve.

As is known, multiple factors including physical and nutritional status influence surgical success. Preoperative education, carbohydrate loading, multimodal analgesia, mobilization, and early and correct nutrition collectively contribute to inflammatory response attenuation, as well as bone and muscular metabolism modulation, avoiding body protein wasting, preserving physical function [7], enhancing insulin sensitivity before surgery, and minimizing potential surgical complications [5]. Thus, a synergistic approach with prehabilitation programs and dietary supplementation could both significantly impact the success of the surgery and maximize surgical treatment outcomes [4].

Exercise and nutritional management are crucial for the prevention of sarcopenia onset and progression, especially in the presurgical period. Although significant advancements have been made in understanding the molecular and cellular regulation of muscle protein metabolism, as well as the identification of sarcopenia as a disease [10] in recent years, approved drugs for sarcopenia treatment are lacking [11]. Similarly, prehabilitation protocols before knee arthroplasty are missing. The patients’ need to comply with prescribed therapeutic regimens, coupled with the intricate diagnosis of sarcopenia, represent a challenge for multidisciplinary teams to select the appropriate therapeutic pathway [12,13,14]. Despite ongoing research to try to develop a clinical algorithm able to face the sarcopenia clinical complexity, both coordinated care and collaboration among healthcare professionals are strongly required.

The aim of this research was to innovatively explore whether the preoperative period could be the appropriate timeframe to empower surgical resilience through the enhancement of functional reserve in older sarcopenic patients who must undergo knee arthroplasty. Moreover, it explored the muscular fiber preparation feasibility, through dietary supplementation and prehabilitation. The final goal was that this multidimensional and multiprofessional surgical preparation should pursue not only the macroscopic functional improvements usually associated with prehabilitation exercises, but also the muscle architectural enhancements promoted by nutritional supplementation.

## 2. Methods

This review followed the Scale for the Assessment of Narrative Review Articles (SANRA) quality criteria [15] and was conducted between December 2023 and February 2024 (Table 1). The current literature concerning the effectiveness of prehabilitation and dietary supplementation before knee arthroplasty in sarcopenic elderly people was reviewed. A comprehensive search was conducted on the PubMed, Scopus, and Web of Science databases in order to identify relevant published studies without any publication date restriction. The search strategy included keywords related to “preoperative exercise” OR “prehabilitation” AND “dietary supplementation” OR “dietary supplements” associated with “knee arthroplasty” AND “sarcopenia”. This search strategy allowed us to yield 469 articles from the Web of Science using the complete research string and 72 articles from the Web of Science, 37 from PubMed, and 1 from Scopus excluding “sarcopenia” as a keyword. Additionally, 5 articles were identified using Mesh of PubMed with the string “preoperative exercise” AND “dietary supplementation”. Furthermore, the included article references were examined to ensure comprehensive coverage of the relevant literature.

The included studies met the following criteria: (1) investigation of prehabilitation and/or dietary supplementation interventions before knee arthroplasty; (2) inclusion of human participants aged 65 years and older; (3) reporting relevant outcomes such as functional status, postoperative complications, and patient-reported outcomes; and (4) articles written in English. Reviews, case reports, letters, or conference abstracts were excluded, as well as studies focusing solely on rehabilitation post-knee arthroplasty without preoperative interventions.

Data extraction and synthesis were performed using a qualitative method. Two independent reviewers (RC and FP) screened the titles and abstracts of the retrieved studies to identify potentially relevant articles. After that, full texts were assessed for eligibility based on the inclusion and exclusion criteria. Extracted information included the study characteristics (e.g., author, publication year, study design), participant demographics, intervention details (e.g., type of prehabilitation, dietary supplementation regimen), outcomes assessed, and main findings. Collected data heterogeneity, different types of prehabilitation or dietary supplementation performed, patients’ comorbidity differences, and the limited number of studies precluded the conduction of a systematic qualitative and quantitative data analysis.

## 3. Results

A total of 12 randomized controlled trials (RCTs) concerning nutritional supplementation before knee arthroplasty were identified (Table 1), and 26 clinical trials plus 1 prospective observational study (Table 2) focused on prehabilitation preceding knee arthroplasty. Interestingly, no studies about the combined use of prehabilitation and dietary supplementation were found, and unexpectedly, adding “sarcopenia” to the search string generated a reduction in the available studies. This finding highlights a significant aspect of clinical practice: the uncommon consideration of sarcopenia as a risk factor for surgical outcomes. Therefore, this risk underestimation emphasizes the importance of sarcopenia diagnosis and treatment in order to improve functional outcomes and prevent frailty progression in these patients.

In the current state, there are no standardized prehabilitation and/or dietary supplementation strategies specifically tailored for patients with sarcopenia [4]. In addition, the type, amount, and duration of training, and/or nutritional supplementation vary across studies. As is known, preoperative pain, baseline physical functioning, and medical, psychosocial and demographic variables can predict functional outcomes in patients undergoing total knee replacement. Marked functional limitations, severe pain, low mental health scales scores, and other comorbidities are more likely related to worse 1- and 2-year postoperative outcomes [29]. In particular, and as prognostic factors, Sharma L. et al. found that baseline demographic variables accounted for 4% of the variance in functional outcomes after total knee replacement; in addition, psychosocial variables (motivation, role of emotional functioning and social functioning) accounted for 19%, medical variables (previous reconstruction, multimorbidity, body mass index, body pain) accounted for 2%, and physical function accounted for 2% [30]. Furthermore, preoperative quadriceps strength was found to be a strong predictor of functional performance 1 year [31] and 2 years [32] after knee arthroplasty. If it is possible to predict surgical outcomes based on a patient’s functional characteristics, it seems reasonable that enhancing the presurgical condition through prehabilitation and supplemental nutrition could lead to improvements in postoperative function. Therefore, the development of specific preoperative training programs and appropriate dietary supplementation seems to be crucial for muscle performance and architecture improvements, respectively.

The vitamin D level assessment prior to knee arthroplasty reveals that its preoperative deficiency negatively impacts knee arthroplasty outcomes. However, postoperative vitamin D supplementation results in similar functional recovery in subjects with or without either deficiency after 3 months from the intervention [33]. In their recent work [34], Kong et al. discussed the critical role of vitamin D in bone health and recovery from orthopedic procedures. They found that an adequate vitamin D level is essential for optimal musculoskeletal function and surgical recovery. Hegde V. and colleagues investigated the impact of preoperative vitamin D deficiency on postoperative complication rates in patients undergoing TKA, using data from the Humana administrative claims registry and measuring vitamin D levels within 90 days before surgery [35]. Patients with a deficiency (25OHD < 20 ng/mL) prior to TKA exhibited significantly higher rates of postoperative complications, including infections requiring surgical intervention and cardiovascular events. These findings highlight the importance of vitamin D deficiency as a modifiable surgical risk factor and suggest that addressing this condition in the preoperative phase could potentially improve surgical outcomes [35].

As is shown in Table 2, essential amino acids (EAAs), iron, vitamin D, adenosine triphosphate, and glucosamine sulphate were proposed as nutritional supplements in knee arthroplasty.

Table 3 defined the prehabilitation strategies extrapolated by the current literature to improve patients’ functional capacity, fortify their ability to withstand the upcoming surgical stressors, and enhance postoperative outcomes.

Table 4 merges information from the above-mentioned tables, in order to provide a quick overview of the preoperative treatment options proposed by the current literature.

## 4. Discussion

To the best of our knowledge, the present study is the first to explore the influence of sarcopenia on knee prosthesis surgery, and how nutritional adjustments and prehabilitation can contribute to improving surgical outcomes. This paper aims to delve into the intricate relationship between knee osteoarthritis, sarcopenia, and the pivotal role of nutraceuticals and prehabilitation as an essential phase preceding orthopedic surgery.

The combination of dietary intervention and exercise is likely to be key in preventing and treating sarcopenia, as described for middle-aged and older females [61].

Total knee arthroplasty is highly effective for treating gonarthrosis, but it can lead to muscle atrophy and long-term deficits, especially in osteoporotic or sarcopenic elderly people.

### 4.1. Sarcopenia in Elderly Individuals with Knee Osteoarthritis

Aging is a complex process involving various changes in body composition, endocrine changes (an imbalance between anabolic and catabolic hormones), and excessive inflammatory responses, prominently featuring the decline in skeletal muscle mass and strength. This gradual reduction in muscle mass is often accompanied by a decrease in muscle strength and function, resulting in reduced mobility, heightened fragility, and a loss of independence among older individuals. The multifaceted nature of sarcopenia, characterized by this physical decline, is attributed to factors such as a sedentary lifestyle, inadequate nutrition, chronic inflammation, and neurological alterations [62]. All older individuals are encouraged to engage in daily physical activity or exercise, encompassing both resistance training and aerobic exercise, for as long as possible [63].

Knee osteoarthritis is frequently linked to reduced physical activity due to joint pain and limited mobility. In this context, elderly people with knee osteoarthritis are prone to developing sarcopenia—a gradual deterioration of muscle mass and strength. This bidirectional relationship between knee osteoarthritis and sarcopenia can further compromise the overall quality of life especially in elderly individuals [3].

### 4.2. Metabolic and Nutritional Changes

The intricate and multifaceted pathophysiology of sarcopenia poses a significant challenge in identifying a specific biomarker that could conveniently reflect both muscle mass and function debilitation. Commonly referenced markers typically indicate the level of inflammation and nutritional status (such as hemoglobin, albumin, C-reactive protein, Interleukin-6, and tumor necrosis factor), oxidative stress (protein carbonylation, oxidized low-density lipoproteins), and hormonal anabolic status (total testosterone, insulin-like growth factor 1 (IGF-1), dehydroepiandrosterone (DHEA), vitamin D). However, none of these markers is exclusive to sarcopenia [64], but they express the disorders and the needs of sarcopenic patients.

Indeed, the decreased secretion of crucial hormones like growth hormone, IGF-I, testosterone, and estradiol, and the involvement of other aging-associated hormones such as DHEA, thyroid hormones, and vitamin D, are mainly related to age and can be associated with the development of sarcopenia [62]. 

Sarcopenia correlates with factors such as aging, mostly with a lower body mass index (BMI), and reduced dietary protein and vitamin D intake [65]. 

Moreover, inflammation, insulin resistance, and physical inactivity contribute to fat deposition, anabolic resistance, and lipotoxicity within the sarcopenic muscle [64].

Furthermore, despite a similar total energy intake, sarcopenic patients consume less protein per kilogram of body weight and have a lower intake of key micronutrients, including vitamin D, vitamin B12, magnesium, phosphorus, and selenium. Thus, the concentration of these nutrients could be lower in sarcopenic patients. In particular, in the study of Verlaan et al. [66], for similar energy intakes, the sarcopenic group exhibited a lower consumption of protein/kg of body weight (−6%), vitamin D (−38%), vitamin B12 (−22%), magnesium (−6%), phosphorus (−5%), and selenium (−2%) (all *p* < 0.05) compared to the non-sarcopenic controls. Specifically, the serum concentration of vitamin B12 was 15% lower in the sarcopenic group, with all other nutrient concentrations showing similar levels between the two groups [66].

As a result, an effective management strategy for sarcopenia should provide all the necessary requirements to positively influence energy homeostasis and body composition. The maintenance of a healthy diet allows people to engage in sufficient physical activity. Integrating a nutritional plan that addresses nutrient quality and intake, the use of anti-inflammatory and anabolic medications, and specific rehabilitation training should be proposed as a multimodal approach [64].

### 4.3. Dietary Supplementation against Sarcopenia in Knee Osteoarthritis 

A range of dietary supplements provides specific health benefits in the context of sarcopenia and knee osteoarthritis. A review provides the plans of the ESCEO working group meeting held on 8 September 2016, for adopting ‘healthier’ dietary patterns in older age, and ensuring adequate intake of protein, vitamin D, antioxidant nutrients, and long-chain polyunsaturated fatty acids. Notably, the review underscores the substantial evidence supporting the roles of dietary protein and physical activity as crucial stimuli for muscle protein synthesis [67].

For healthy older individuals, the dietary consumption of at least 1.0–1.2 g of protein per kilogram of body weight per day is recommended. On the other hand, individuals with severe illness or injury, who are malnourished, or who are at risk of malnutrition due to acute or chronic illnesses require an even higher protein intake with older individuals requiring 1.2–1.5 g of protein per kilogram of body weight per day [63]. Essential amino acid (EAA) supplementation before and after knee arthroplasty mitigates muscle atrophy and positively influences markers of inflammation [23]. It was shown that 20 g of EAAs twice daily between meals for 1 week before and 2 weeks after knee arthroplasty attenuated muscle atrophy and accelerated the return of functional mobility in older adults [20]. The use of β-hydroxy β-methylbutyric acid (HMB), a bioactive metabolite of leucine, is suggested in combination with arginine, an amino acid that contributes to nitric oxide production, enhancing blood flow and nutrient delivery to muscles, and supporting overall muscle health [68]. HMB is also recommended in combination with arginine and lysine [68] or with arginine and glutamine [24] to suppress the expected loss of muscle strength occurring after knee arthroplasty. Recent studies suggest that the strength of the rectus femoris muscle area and quadriceps muscle is significantly improved from baseline after 9 g daily of EEAs (threonine, lysine, isoleucine, valine, methionine, tryptophan, phenylalanine, leucine, histidine, arginine, and glycine). EEA intake has been recommended to be continued from 1 week prior to surgery to 2 weeks after, for four weeks after surgery [25], and maintained for 2 years after knee arthroplasty [26] by different studies.

Leucine combined supplementation including vitamin D exhibited a significant benefit for muscle strength and performance including handgrip strength and gait speed in older adults [69].

Earlier martial status optimization after knee arthroplasty is very important [16]. Intra-operative [28] and postoperative [19] administration of IV iron supplements during knee arthroplasty is crucial for preventing anemia and reducing the need for a blood transfusion during hospitalization. However, 200 mg of ferrous sulphate, containing 65 mg of elemental iron, did not seem to be effective [22].

Additionally, administering a daily dose of recombinant human erythropoietin (rhEPO) combined with iron supplements three days before total knee arthroplasty has been shown to significantly reduce perioperative blood loss and improve postoperative hemoglobin levels. This approach does not significantly increase the risk of complications compared to administering rhEPO on the day of surgery or using iron supplements alone. Therefore, preoperative daily treatment with rhEPO may represent a more effective blood-saving protocol for surgical procedures [18].

Vitamin D, essential for bone health, plays a vital role in calcium absorption. Furthermore, recent research has highlighted the significant extra-skeletal effects of vitamin D, particularly its role in enhancing muscle function. This is especially relevant in the context of postoperative recovery from knee surgery. Vitamin D interacts with muscle tissue through vitamin D receptors (VDRs) present in muscle cells that can influence muscle cell proliferation, differentiation, and function. Adequate levels of vitamin D have been associated with improved muscle strength and reduced muscle atrophy, which is crucial for the recovery and rehabilitation process following knee surgery. Specifically, vitamin D aids in restoring quadriceps strength, thereby enhancing overall recovery outcomes and functional performance in patients undergoing knee surgery. 

Administered as drops, molly capsules, and oro-dispersible films (ODF), vitamin D is gradually gaining great importance for the extra-skeletal effects that are becoming increasingly evident, not only in sports, but also in pre-operative settings, as proposed. Nutritional therapy for sarcopenia, incorporating 20 g of whey proteins and 800 IU of vitamin D twice a day, has been shown to enhance lower-limb strength [1]. In a 13-week intervention involving a vitamin-D- and leucine-enriched whey protein oral nutritional supplement, improvements in muscle mass and lower-extremity function were observed among sarcopenic older adults [70]. Consistent beneficial effects on strength and balance have been demonstrated with supplemental vitamin D at daily doses of 800 to 1,000 IU [71]. However, supplementation with 50,000 IU vitamin D3 on the day of knee arthroplasty failed to demonstrate significant differences in functional outcomes, assessed by the Knee Society Score and Timed Up and Go Test [27].

Adenosine triphosphate (ATP) is a nutritional supplement to improve a person’s ability of exercise [72]. Oral supplementation with ATP could improve recovery from knee arthroplasty [21]. 

Patients with knee OA who had received oral glucosamine sulphate 1500 mg once-a-day for at least 12 months and up to 3 years had a lower incidence of a total joint replacement [17,73,74]. In particular, patients who were previously taking glucosamine sulphate experienced a 57% decrease in the risk of undergoing a total knee replacement [73]. Indeed, glucosamine sulphate seemed to prevent joint structure changes in patients with knee osteoarthritis, as assessed by radiographic joint space narrowing, with a significant improvement in pain and functional limitation [17,73,74].

### 4.4. Dietary Supplementation to Improve Muscle Mass 

Several substances have been proposed to increase strength and should therefore be considered to counteract arthrogenic inhibition. These compounds could be useful in attenuating the strength deficits related to potential sarcopenia, arthritis-related immobility, and surgical pain.

Probiotics, known for their ability to promote the production of metabolites such as short-chain fatty acids, secondary bile acids, and certain amino acids, have demonstrated potential in modulating muscle function [75]. Ni et al. discovered that supplementation with L. casei LC122 and B. longum BL986 for 12 weeks in aged mice led to a significant increase in forelimb grip strength [76]. Consistently, aged SAMP8 mice supplemented with Lactobacillus paracasei PS23 for 12 weeks exhibited attenuated age-related decreases in grip force and displayed higher mitochondrial function compared to controls [77]. These data open new perspectives on the connection between muscle strength and the gut microbiota. However, while research in rodents has shown promising results in limiting sarcopenia and cachexia or enhancing health performances, particularly with lactic acid bacteria and bifidobacteria strains, studies in humans have been limited [75].

Challenges such as the scarcity of research, variability in study populations, and difficulty in accurately measuring muscle mass and function have hindered the identification of specific strains that optimize muscle health [75]. 

The use of omega-3 fatty acids can be valuable in geriatric populations because they promote skeletal muscle anabolism, mitigate muscle atrophy, and facilitate recovery from surgery and minimize subsequent bedrest or inactivity. However, the efficacy of omega-3 fatty acid intake in enhancing skeletal muscle anabolism may vary depending on several factors, including daily protein intake, measurement techniques, as well as the age and metabolic status of participants. Further studies are needed to address these variables and provide more conclusive evidence [78]. The study by Huang et al. proposes a supplement of 2 g/day of omega-3 fatty acids to obtain positive outcomes in sarcopenia-related performances among the elderly and underlined that a long period of omega-3 fatty acids supplementation may improve walking speed [79].

A supplement combination, including protein, leucine, vitamin D, and n-3 polyunsaturated fats (PUFAs) may offer additional advantages and could serve as a preventative measure against sarcopenia and functional deterioration, particularly in the preoperative window of orthopedic surgery.

### 4.5. Prehabilitation: An Essential Step before Orthopedic Surgery

Physiological changes in elderly people or comorbidities occurring prior to surgery may impact postoperative outcomes. Therefore, there is the need for careful adjustments or interventions before the surgical procedure to optimize and improve the overall result for the patient. In the management of knee osteoarthritis, rehabilitation becomes essential to preserve and enhance the surrounding musculature, thereby mitigating the negative impact of sarcopenia.

It is well-established that rehabilitation and adapted physical exercise play a fundamental role in combating the effects of sarcopenia [2]. In particular, high-risk populations like frail or sarcopenic patients stand to gain greater benefits from such exercise prehabilitation efforts [4].

Prehabilitation aims to enhance muscle strength and endurance, preventing the progression of sarcopenia and promoting a quicker postoperative recovery. Additionally, the main goal of prehabilitation is to improve the physiological reserve and fitness, targeting modifiable risk factors to mitigate surgical complications [4]. Moreover, addressing sarcopenia before orthopedic surgery can decrease the risk of postoperative complications, such as muscle atrophy and infections, contributing to a faster and more effective recovery. Lastly, adequate prehabilitation can improve the quality of life, allowing for greater independence and autonomy in daily activities.

Due to pain-related limitations in mobility and exercise, patients undergoing total knee arthroplasty often experience reduced muscular function before surgery [79]. Thus, although a benefit of preoperative training prior to knee surgery is likely, there is no consensus on the optimal content (criteria-based program), supervision (one-on-one guidance or self-administered training), and general setting of preoperative training [51]. However, the literature confirmed that multimodal prehabilitation enhanced pre- and postoperative functional capacity in elective surgical patients undergoing knee procedures [49,51,80,81]. Indeed, prehabilitation assumes a pivotal role in the treatment path for elderly individuals with sarcopenia and knee osteoarthritis, even reducing the need for knee arthroplasty by 20% [56]. 

Resistance training, flexibility, and aerobic training for 4–8 weeks improve knee pain, functional ability and quadriceps strength [36,37]. Eight weeks of aquatic exercises of strength and resistance resulted in improved functional outcomes as well as improved depression and cognition [39]. Just three days of aerobic and straitening exercises before surgery can significantly reduce pain, improve the functional state of knee joints, shorten the time to get out of bed for the first time after surgery, and reduce the medical care services level [41]. Endurance training combined with proprioceptive neuromuscular facilitation (PNF), defined as the contract–relax–antagonist–contract (CRAC) program, for 3–4 weeks before surgery improved the level of physical activity [49]. 

Blood-Flow-Restriction Exercises (BFRE) use specialized tourniquets during exercise to restrict venous and reduce arterial blood flow, increase metabolic stress, and improve muscle function and QoL before total knee arthroplasty [51]. BFRE prehabilitation for 6 weeks seems to reduce perceived pain and increase muscle mass and strength significantly before elective knee surgery, with supportive effects on pain perception and quality of life [51]. Furthermore, BFRE prehabilitation shows a positive influence on the postoperative improvement of skeletal muscle mass after 3 weeks [51]. 

High-intensity strength training 8 weeks before surgery reduces pain and improves lower-limb muscle strength, ROM, and functional task performance, resulting in a reduced length of hospitalization and a faster physical and functional recovery after total knee arthroplasty [52]. Short-term high-intensity resistance training for 4 weeks before surgery helps to maintain knee ROM up to one year post-surgery [54]. However, according to Beaupre et al. [48], a program of strengthening 4 weeks before surgery does not significantly improve ROM, strength, pain, function, and quality of life in patients undergoing arthroplasty. 

Balance training is essential to reduce the risk of falls. Training for 4 weeks before surgery seems to have prolonged effects for 6 months after surgery [55].

Not only physiotherapist-guided rehabilitation, but also self-guided rehabilitation at home seems to be effective [40]. Home rehabilitation together with an education program and muscle strength and flexibility exercises for 4 and 6 weeks, respectively, before surgery can reduce the days lost and increase cost savings [42], in addition to increasing mobility and knee function [43,44]. Even just 3 weeks of a home quadriceps-strengthening exercise program can improve pain, strength, and quality of life in the 3 months after surgery [57]. Moreover, 3 months of tele-prehabilitation with strengthening and proprioceptive exercises generated good satisfaction [47] and this can improve the adherence to treatment. However, Soeter et al. did not find effects in terms of WOMAC and days of hospitalization after a 2-week home exercise program before surgery [58].

However, it should be underlined that patients could not perceive the effective improvement in strength, functional performance, and pain [53]. We can also hypothesize that significant results were not always recorded. Villadsen et al. [50] and Huber et al. [38] did not find significant outcomes after a specific neuromuscular exercise program of 8 and 4 weeks before surgery, respectively, in the recovery of function, autonomy in activities of daily living, and pain. Mat Eil-Ismail et al. [46] showed that 6-week preoperative physiotherapy showed no significant impact on short-term functional outcomes (KOOS subscales) and ROM. Soni et al. [59] showed that the use of combined acupuncture and physiotherapy in the treatment of patients with moderate-to-severe knee OA preoperatively did not improve patient outcomes postoperatively. Furthermore, Williamson et al. [60] did not show significant results after 7 weeks of prehabilitation, but they found a short-term reduction in the Oxford Knee Score (OKS) questionnaire after acupuncture. Accordingly, Cavill et al. [45] showed partial positive results; in fact, prehabilitation improved knee flexion, without affecting functional mobility or quality of life. 

The limitation of the available literature on the topic, especially regarding prehabilitation and nutritional supplementation specifically in sarcopenic patients undergoing knee arthroplasty, might restrict the depth of analysis and synthesis. The variability in interventions (dosage, timing, duration), and outcome measures across the included studies could pose challenges in synthesizing findings and drawing conclusive insights. 

The diagnosis of sarcopenia, particularly in its early stages, poses challenges. Indeed, sarcopenia itself is a multifaceted condition influenced by various factors such as age, multimorbidity, and physical activity levels, which might introduce complexity in interpreting the effectiveness of prehabilitation and nutritional supplementation interventions. Furthermore, limited longitudinal data on the long-term effects of prehabilitation and nutritional supplementation in sarcopenic patients undergoing knee arthroplasty might hinder the ability to assess sustained benefits or adverse outcomes over time. 

Moreover, the decision to proceed with knee arthroplasty in patients with incongruence between clinical symptoms and radiological images further complicates the therapeutic pathway. 

## 5. Limitations

The limited available literature, especially that inherently concerning prehabilitation and dietary supplementation in sarcopenic older patients undergoing knee arthroplasty, restricted the magnitude of the topic analysis and synthesis. The heterogeneity in interventions and outcome selection across the included studies posed challenges in synthesizing findings and drawing conclusive insights. The multifaceted nature of sarcopenia, with its known challenging diagnosis and complex pathophysiology, influenced by age, multimorbidity, and physical activity degree, makes it difficult to interpret the effectiveness of prehabilitation and dietary supplementation in this particular preoperative scenario. Furthermore, the lack of longitudinal data regarding the long-term effects of prehabilitation and dietary supplementation in sarcopenic older patients undergoing knee arthroplasty reduces the ability to evaluate the benefits or adverse postoperative outcomes over time. Moreover, the frequent surgical decision to proceed to knee arthroplasty despite the incongruence between clinical symptoms and radiological findings further complicates the therapeutic pathway. 

Furthermore, a limitation of this review is not only the inability to conduct a systematic review and meta-analysis, but also the failure to address numerous important aspects in a clinical pathway as heterogeneous as that of the osteoarthritic patient with a knee prosthesis. While pre-supplementation, rehabilitation, and postoperative recovery are discussed, the review does not take into account the surgical technique and type of prosthetic implant, which can significantly influence patient outcomes. The surgical technique, involving the resection of muscle fibers and joint capsule, followed by suturing after implant placement, can alter the alignment of fibers and the force vector generated during contraction. This highlights the heterogeneity in reconstruction, which, while aiming to be as anatomical as possible, can never be fully anatomical. Furthermore, prosthetic implants are based on completely different biomechanical rationales, with varying surface congruence, degrees of freedom, and coupling mechanisms. These factors will need to be considered when clinical trials are initiated.

In conclusion, diagnostic uncertainty and surgical decisional complexity introduce limitations in designing clinical studies used to derive evidence-based recommendations regarding prehabilitation and preoperative nutritional supplementation, especially in populations with complex needs such as sarcopenic older people.

## 6. Conclusions

Addressing the complex links between knee osteoarthritis and sarcopenia in older individuals undergoing orthopedic interventions requires a multidimensional approach. Prehabilitation and dietary supplementation are essential for surgical resilience, in terms of muscle function preservation, recovery acceleration, and overall quality of life enhancement.

Prehabilitation emerges as a crucial preliminary step, allowing the optimization of orthopedic surgery outcomes. Nutraceutical integration, as part of a comprehensive care plan, could have a synergic effect in achieving prehabilitation goals, such as sarcopenic older people.

Further research is required, especially in such a specific population as sarcopenic and/or malnourished older people, to promote the development of personalized nutritional strategies. It is essential for older people to evaluate the effectiveness of nutraceuticals based on clinical and functional outcomes such as frailty, nutritional status, and the degree of sarcopenia prior to supplementation. 

A comprehensive approach requires intensive collaboration among specialists in rehabilitation and exercise physiology, orthopedics, internal medicine, geriatrics, nutrition, and basic science, in order to advance knowledge and the sharable consensus concerning this critical and evolutive field of perioperative care, as well as to face the challenges posed by the multifaced interaction between knee osteoarthritis and sarcopenia in older people.

## Figures and Tables

**Table 1 nutrients-16-03462-t001:** SANRA Evaluation.

Study	Justification of Importance (0–2)	Concrete Aims/Questions (0–2)	Literature Search Description (0–2)	Referencing (0–2)	Scientific Reasoning (0–2)	Appropriate Data Presentation (0–2)	Total Score (0–12)
Briguglio et al., 2020 [16]	2	2	0	2	2	2	10
Bruyere et al., 2008 [17]	2	2	1	2	1	1	9
Cao et al., 2020 [18]	2	2	0	2	2	2	10
Choi et al., 2022 [19]	2	2	0	2	2	2	10
Dreyer et al., 2013 [20]	2	2	0	2	2	2	10
Long and Zhang, 2014 [21]	2	2	0	2	2	2	10
Mundy et al., 2005 [22]	1	1	0	1	2	1	6
Muyskens et al., 2019 [23]	2	1	0	1	1	1	6
Nishizaki et al., 2015 [24]	2	2	0	2	2	2	10
Ueyama et al., 2020 [25]	2	2	0	2	2	2	10
Ueyama et al., 2023 [26]	2	2	0	2	2	2	10
Weintraub et al., 2023 [27]	2	1	0	1	1	1	6
Yoo et al., 2021 [28]	2	2	0	2	2	2	10

0 = low; 1 = moderate; 2 = high.

**Table 2 nutrients-16-03462-t002:** Characteristics of the studies on nutritional supplementation in knee arthroplasty.

Supplementation	Authors	Study Design	Sample Size, Age	Outcomes	Results	Conclusions
**Iron**						
Sucrosomial ferric pyrophosphate (30 mg) plus L-ascorbic acid (70 mg)	Briguglio et al. [16]	RCT	N = 73, 67.3 ± 8.6. Intervention group: N = 37, Control group: N= 36	Anemia	Older patients with no support lost −2.8 ± 5.1%, while the intervention group gained +0.7 ± 4.6% circulating hemoglobin from baseline (*p* = 0.019)	After 30 days of oral iron plus L-ascorbic acid therapy, no significant changes in the martial status were observed after treatment.
Ferric carboxymaltose 1000 mg (body weight ≥ 50 kg) Ferric carboxymaltose 500 mg (body weight < 50 kg) For 1 day before and 3 days after surgery	Choi et al., 2022 [19]	RCT	110 pt, Intervention group: N = 54, 71.4 ± 5.7 Control group: N = 55, 71.8 ± 6.2	Hb and iron response, QoL	The FCM group demonstrated a significantly greater number of Hb responders (*p* < 0.001) and a higher Hb level (*p* = 0.008) at 2 weeks postoperatively.	In postoperative anemia, a single infusion of 5000 and 1000 mg ev of ferric carboxymaltose increases the Hb response and improved Hb and iron metabolism variables. However, intervention did not affect the transfusion rate or QOL.
Ferrous sulphate (200 mg, containing 65 mg of elemental iron) 3 times daily for 3 weeks	Mundy et al., 2005 [22]	RCT	31 pt, Intervention group: N = 18, 67.8 ± 10.5 Control group: N = 13, 67.0 ± 9.4	Hemoglobin and absolute reticulocyte count	Administration of iron supplements after elective total hip or total knee arthroplasty does not appear to be worthwhile.	Administration of iron supplements after elective total hip or total knee arthroplasty does not appear to be worthwhile.
Iron isomaltoside administered 30 min during surgical wound closure.	Yoo et al., 2021 [28]	RCT	89 pt Intervention group N = 44 71 ± 6 Control group N = 45 70 ± 7	Hb	The incidence of anemia at 30 days after surgery was significantly lower in the treatment group (*p* = 0.008)	The intra-operative administration of iron isomaltoside effectively prevents postoperative anemia
Daily doses of rhEPO combined with iron supplement	Cao et al., 2020 [18]	RCT	102 pt Group A: rhEPO + iron (3 days before surgery), Group B: rhEPO + iron (day of surgery), Group C: iron alone	Hb, blood loss, reticulocyte levels, complications	Patients in Group A had significantly lower total blood loss than Groups B and C (A vs. B: *p* = 0.010; A vs. C: *p* < 0.001). Group A patients had significantly smaller Hb decline than Group C on the third and fifth postoperative days (*p* = 0.049, *p* = 0.037), as well as than Group B on the fifth postoperative day (*p* = 0.048)	Daily dose of rhEPO combined with iron supplement administered 3 days before TKA procedures could significantly decrease perioperative blood loss and improve postoperative Hb levels, without significantly elevating risks of complications
**Glucosamine sulphate**						
Oral glucosamine sulphate 1500 mg once-a-day for at least 12 months and up to 3 years.	Bruyere et al., 2008 [17]	RCT	275 pt, Intervention group: N = 144, 62.9 ± 7.6 Control group: N = 131, 63.6 ± 6.6	Number of knee arthroplasties	A significantly decreased (*p* = 0.026) cumulative incidence of total knee replacements in patients who had received glucosamine sulphate.	Treatment of knee OA with glucosamine sulphate for at least 12 months and up to 3 years may prevent surgery
**Essential Amino Acids**						
20 g of EAA Twice daily between meals for 1 week before and 2 weeks after surgery.	Dreyer et al., 2013 [20]	RCT	28 pt, Intervention group: N = 16, 68 ± 5 Control group: N = 12, 70 ± 5	Muscle atrophy, muscle strength, and functional mobility	Patients receiving placebo exhibited greater quadriceps muscle atrophy, 2 weeks (*p* = 0.036) and 6 weeks after surgery (*p* = 0.001)	EAA treatment attenuated muscle atrophy and accelerated the return of functional mobility in older adults following surgery
EAA 20 g of EAA twice-daily, for 7 days before and for 6 weeks after surgery	Muyskens et al., 2019 [23]	RCT	N = 41	A biopsy during surgery, and two additional biopsies at either 1 or 2 weeks after surgery to study satellite cells and other key histological markers of inflammation, recovery, and fibrosis.		
L-arginine (Arg; 14,000 mg) and L-glutamine (Gln; 14,000 mg) (HMB/Arg/Gln), Beta-hydroxy beta-methylbutyrate (HMB; 2400 mg), (158 kcal of energy) for 5 days before the surgery and for 28 days after the surgery. Patients fasted on the day of surgery.	Nishizaki et al., 2015 [24]	RCT	N = 32	Body weight, bilateral knee extension strength, rectus femoris cross-sectional area	The maximal quadriceps strength was 1.1 ± 0.62 Nm/Kg before surgery and 0.7 ± 0.9 Nm/Kg after surgery 14 days in the control group (*p* = 0.02), and 1.1 ± 0.3 Nm/Kg before surgery and 0.9 ± 0.4 Nm/Kg after surgery 14 days in the HMB/Arg/Gln group.	Consuming HMB/Arg/Gln supplementation may suppress the loss of muscle strength after TKA. Intervention with exercise and nutrition appears to enable patients to maintain their quadriceps strength.
Isoleucine (603 mg, 6.7%), leucine (684 mg, 7.6%), lysine (756 mg, 8.4%), methionine (603 mg, 6.7%), phenylalanine (405 mg, 4.5%), threonine (405 mg, 4.5%), tryptophan (207 mg, 2.3%), valine (603 mg, 6.7%), arginine (630 mg, 7%), histidine (315 mg, 3.5%), and starch (1089 mg, 12.1%) From 1 week prior to surgery until 2 weeks after it. 3 times daily (after every meal) for a total of 9 g/day.	Ueyama et al., 2020 [25]	RCT	60 pt Intervention group N = 30 75.9 Control group N = 30 75.8	Rectus femoris muscle area	Improvement of VAS (*p* = 0.038), albumin level (*p* = 0.009), quadriceps area (*p* = 0.026), muscle diameter (*p* = 0.029) after 4 weeks from surgery	Perioperative essential amino acid supplementation prevents rectus femoris muscle atrophy and accelerates early functional recovery after surgery
Threonine (405 mg, 4.5%), lysine (756 mg, 8.4%), isoleucine (603 mg, 6.7%), valine (603 mg, 6.7%), methionine (603 mg, 6.7%), tryptophan (207 mg, 2.3%), phenylalanine (405 mg, 4.5%), leucine (684 mg, 7.6%), histidine (315 mg, 3.5%), arginine (630 mg, 7%), and glycine (1089 mg, 12.1%); the remainder was starch (2700 mg, 30%).	Ueyama et al., 2023 [26]	RCT	52 pt Intervention group N = 26 76.4 ± 8.3 Control group N = 26 75.2 ± 5.5	Rectus femoris muscle area	Improvement in rectus muscle area (*p* = 0.02, *p* = 0.01), diameter (*p* = 0.009) after 1 year and 2 years	Perioperative EAA supplementation contributes to the recovery of rectus femoris muscle volume and quadriceps muscle strength in the 2 years after surgery
**Vitamin D3**						
50,000 international units vitamin D3 on the morning of surgery	Weintraub et al., 2023 [27]	RCT	107 pt Intervention group N = 57 63.7 ± 9.5 Control group N = 50 64.5 ± 63.7	KSS, TUG	There was no difference in improvement in KSS at 3 weeks (*p* = 0.6) or 6 weeks (*p* = 0.5) from baseline. There was no difference in change in TUGT at 3 weeks (*p* = 0.6) or 6 weeks (*p* = 0.6) from baseline.	Supplementation with vitamin D3 on the day of surgery failed to demonstrate statistically significant differences in functional KSS, TUGT times, or complications in the early postoperative period compared to placebo.
**Adenosine 5′-triphosphate supplementation**						
ATP 20 mg ATP for 4 weeks	Long and Zhang, 2014 [21]	RCT	232 pt, Intervention group: N = 119, 60.1 ± 4.5 Control group: N = 113, 58.9 ± 5.2	Quadriceps strength, pain scores	Reduction in length of hospitalization (*p* = 0.0027) and analgesic consumption (*p* = 0.021)	Oral supplement of ATP could benefit patients recovering from knee arthroplasty

Essential amino acids, EAA; QoL, quality of life; Hb, hemoglobin; Knee Society Score, KSS; Timed Up and Go, TUG; adenosine triphosphate, ATP; recombinant human erythropoietin, rhEPO.

**Table 3 nutrients-16-03462-t003:** Characteristics of the studies on prehabilitation before knee arthroplasty.

Training	Timing: Weeks before Surgery	Results	References
Straitening and resistance training, flexibility, and aerobic training	4–8 weeks	Improvement in knee pain, functional ability, and quadriceps strength	Topp et al., 2009 [36], Swank et al., 2011 [37], Huber et al., 2015 [38]
	8 weeks	Improvement in depression and cognition and quality of life	Kim et al., 2021 [39], Brown et al., 2012 [40]
	3 days	Reduction in pain, improve functional state of knee joints	Zheng et al., 2022 [41]
	4 and 6 weeks	Reduction in the days lost and increase cost savings	Huang et al., 2011 [42]
	6 weeks	Increase in mobility and knee function	Matassi et al., 2012 [43], Jahic et al., 2018 [44]
Strengthening and aerobic program	4 weeks	Improvement in knee range of motion	Cavill et al., 2016 [45]
	6 weeks	Improvement in pain, quality of life, ROM	Mat Eil-Ismail et al., 2016 [46]
Strengthening, aerobic and proprioceptive exercises	12 weeks	Improvement in satisfaction	Doiron-Cadrin et al., 2020 [47]
Program of strengthening	4 weeks	No improvement in ROM, strength, pain, function, and QoL	Beaupre et al., 2004 [48]
Endurance training + proprioceptive neuromuscular facilitation (PNF)	3–4 weeks	Improvement in the level of physical activity	Gränicher et al., 2020 [49]
Neuromuscular and postural exercises	8 weeks	No additional benefits	Villadsen et al., 2013 [50]
Blood-Flow-Restriction Exercises	6 weeks	Reduction in pain; increase in metabolic stress; improvement in muscle function and QoL, muscle mass and strength	Franz et al., 2022 [51]
High-intensity strength training	8 weeks	Reduction in pain and improvement in lower limb muscle strength, ROM, and functional task performance	Calatayud et al., 2017 [52]
	4 weeks	Improvement in ROM	Skoffer et al., 2016 [53], Skoffer et al., 2020 [54]
Balance training	4 weeks	Improvement in balance	Domínguez-Navarro et al., 2021 [55]
Home-based exercises	8–12 weeks	Increase in knee flexion and extension, gait re-education, and home/functional adaptations	Aytekin et al., 2019 [56]
	3 weeks	Decreased pain, improved quadriceps strength, and improved quality of life	Tungtrongjit et al., 2012 [57]
	2 weeks	No effect on days of hospitalization and WOMAC scores	Soeter et al., 2017 [58]
Acupuncture and strength exercises	4 weeks	Acupuncture and physiotherapy preoperatively did not improve patient outcomes postoperatively	Soni et al., 2012 [59]
	6 weeks	Reduction in OKS	Williamson et al., 2007 [60]

Quality of life, QoL; range of motion, ROM; Oxford Knee Score questionnaire, OKS.

**Table 4 nutrients-16-03462-t004:** Flow chart of preoperative supplementation and prehabilitation.

Preoperative Strategies
Nutritional Supplementation	Prehabilitation
**Ferric supplementation**
-Sucrosomial ferric pyrophosphate plus L-ascorbic acid-Ferric carboxymaltose-Ferrous sulphate-Iron isomaltoside-rhEPO + iron	-Straitening, resistance training, flexibility, and aerobic training-Strengthening and aerobic program-Strengthening, aerobic, and proprioceptive exercises
**Glucosamine sulphate**	
	-Program of strengthening
**Essential Amino Acids**	
-L-arginine-L-glutamine-Beta-hydroxy beta-methylbutyrate-Histidine-Isoleucine-Leucine-Lysine-Methionine-Phenylalanine-Starch-Threonine-Tryptophan-Valine	-Endurance training + proprioceptive neuromuscular facilitation (PNF)-Neuromuscular and postural exercises-Blood-Flow-Restriction Exercises-High-intensity strength training-Balance training-Home-based exercises
**Vitamin D3**	-Acupuncture and strength exercises
**Adenosine 5′-triphosphate**

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
