# Peer review of "Optimizing the Preoperative Preparation of Sarcopenic Older People: The Role of Prehabilitation and Nutritional Supplementation before Knee Arthroplasty"

_nutrients, 2024, doi:10.3390/nu16203462_

Round 1
Reviewer 1 Report
Comments and Suggestions for Authors
Dear Authors,
It is a review of literature on basic health problems among older individuals referred for orthopaedic procedures due to degenerative disease of the knee joints and preparation of patients in the pre-operative period, through supplementation and physical rehabilitation. It concerns the significance of physical pre-rehabilitation and supplementation in the period before knee joint endoprosthesis implantation surgery, in order to reduce post-operative complications related to, among others, sarcopenia in seniors. The undertaken topic is also important in the practical dimension. The literature review was conducted in accordance with applicable standards.Minor comments concern, in particular:
1) There is no need to indicate the innovative nature of the study in the purpose; the aim of the work is to review the literature in the subject area (defined by the authors).
2) The term "nutritional support" used by the authors in reference to their work should be replaced with the term "supplementation”, because the review concerns dietary supplements, not nutrition in the sense of a diet. The authors also use the term "dietary intervention", which is broader in meaning than the described supplementation intervention. Also, in the ‘Discussion’ and ‘Conclusions’ sections, the authors write about the impact of changes in diet, while they reviewed literature on the impact of selected dietary supplements (not changes in the way of eating). I suggest verifying the test with greater attention to scientific precision of language.
3) In the ‘Limitations’ section, the authors indicated weaknesses mainly related to the scientific value of the works included in the review; I suggest a broader view, also critical of their work, outlining directions for further research
4) Keywords: dietary supplements
Author Response
Dear Authors,
It is a review of literature on basic health problems among older individuals referred for orthopaedic procedures due to degenerative disease of the knee joints and preparation of patients in the pre-operative period, through supplementation and physical rehabilitation. It concerns the significance of physical pre-rehabilitation and supplementation in the period before knee joint endoprosthesis implantation surgery, in order to reduce post-operative complications related to, among others, sarcopenia in seniors. The undertaken topic is also important in the practical dimension. The literature review was conducted in accordance with applicable standards.
Thank you very much for taking the time to review this manuscript. Please find the detailed responses below and the corresponding revisions in the re-submitted files.
Minor comments concern, in particular:
1) There is no need to indicate the innovative nature of the study in the purpose; the aim of the work is to review the literature in the subject area (defined by the authors).
Response 1: As correctly suggested by the reviewer, the definition of the objective has been revised by removing subjective opinions that could divert attention from the true purpose of the paper, which is to conduct a review of the scientific literature in the established field.The term “Innovatively” was removed from line 116.
2) The term "nutritional support" used by the authors in reference to their work should be replaced with the term "supplementation”, because the review concerns dietary supplements, not nutrition in the sense of a diet. The authors also use the term "dietary intervention", which is broader in meaning than the described supplementation intervention. Also, in the ‘Discussion’ and ‘Conclusions’ sections, the authors write about the impact of changes in diet, while they reviewed literature on the impact of selected dietary supplements (not changes in the way of eating). I suggest verifying the test with greater attention to scientific precision of language.
Response 2: As correctly suggested by the reviewer, the definition of the objective has been revised by removing subjective opinions that could divert attention from the true purpose of the paper, which is to conduct a review of the scientific literature in the established field. We changed “nutritional support” with “Dietary supplements” in the text.
3) In the ‘Limitations’ section, the authors indicated weaknesses mainly related to the scientific value of the works included in the review; I suggest a broader view, also critical of their work, outlining directions for further research
Response 3: We agree with the integration, describing additional limitations that will be useful when designing future clinical trials. Below is the integrated section included in the "Limitations" section of the text.
“The limitation of this review is not only the inability to conduct a systematic review and meta-analysis, but also the failure to address numerous important aspects in a clinical pathway as heterogeneous as that of the osteoarthritic patient with a knee prosthesis. While pre-supplementation, rehabilitation, and post-operative recovery are discussed, the review does not take into account the surgical technique and type of prosthetic implant, which can significantly influence patient outcomes. The surgical technique, involving the resection of muscle fibers and joint capsule, followed by suturing after implant placement, can alter the alignment of fibers and the force vector generated during contraction. This highlights the heterogeneity in reconstruction, which, while aiming to be as anatomical as possible, can never be fully anatomical. Furthermore, prosthetic implants are based on completely different biomechanical rationales, with varying surface congruence, degrees of freedom, and coupling mechanisms. These factors will need to be considered when clinical trials are initiated.”
4) Keywords: dietary supplements
Response 4: we added “dietary supplements” in the list of keywords
Reviewer 2 Report
Comments and Suggestions for Authors
Review of "Optimizing preoperative preparation of sarcopenic older people: the role of prehabilitation and nutritional supplementation before knee arthroplasty" (nutrients-3217422).
This review article focuses on the effect of pre-rehabilitation exercises and supplementary nutritional on improving functional recovery and performance. The concept itself was interesting and important. Unfortunately, however, this reviewer has several questions and comments.
1. Although this is a narrative review, it would be better to indicate how much of the literature has been narrowed down.
2. The numbering of references should be modified. Some references are given only in tables and are difficult to read.
3. In this review, the authors focused on “sarcopenia”, and few studies on sarcopenia were included. From this perspective, the title of this paper was not considered consistent with its content.
4. Furthermore, the target outcome of Tables was differ from the purpose of this study.
5. SANRA score was not presented. This reviewer would like to see a summary of nutritional supplementation and prehabilitation that would be necessary as a NARRATIVE REVIEW without comprehensive search.
Author Response
Comments and Suggestions for Authors
Review of "Optimizing preoperative preparation of sarcopenic older people: the role of prehabilitation and nutritional supplementation before knee arthroplasty" (nutrients-3217422).
This review article focuses on the effect of pre-rehabilitation exercises and supplementary nutritional on improving functional recovery and performance. The concept itself was interesting and important. Unfortunately, however, this reviewer has several questions and comments.
Thank you very much for taking the time to review this manuscript. Please find the detailed responses below and the corresponding revisions in the re-submitted files.
- Although this is a narrative review, it would be better to indicate how much of the literature has been narrowed down.
Response 1: We fully agree and believe that clarifying the points suggested makes this review even more valuable and important, as it draws attention to a clinical phase that is often overlooked and underestimated.
- The numbering of references should be modified. Some references are given only in tables and are difficult to read.
Response 2: Thank you for the suggestion, it is indeed true. We have reorganized the references to provide a clear overview of the scientific sources and to allow readers to easily locate them.
- In this review, the authors focused on “sarcopenia”, and few studies on sarcopenia were included. From this perspective, the title of this paper was not considered consistent with its content.
Response 3: Thank you for the suggestion, the title was changed
- Furthermore, the target outcome of Tables was differ from the purpose of this study.
Response 4: Thank you for the suggestion, it is indeed true. We have reorganized the references to provide a clear overview of the scientific sources and to allow readers to easily locate them.
- SANRA score was not presented. This reviewer would like to see a summary of nutritional supplementation and prehabilitation that would be necessary as a NARRATIVE REVIEW without comprehensive search.
Response 5: we agree with this comment. We added a table on SANRA score (see Material and Methods)

Round 2
Reviewer 2 Report
Comments and Suggestions for Authors
The authors revised well.